# Effects of Non-Invasive Brain Stimulation on Post-Stroke Spasticity: A Systematic Review and Meta-Analysis of Randomized Controlled Trials

**DOI:** 10.3390/brainsci12070836

**Published:** 2022-06-27

**Authors:** Xiaohan Wang, Le Ge, Huijing Hu, Li Yan, Le Li

**Affiliations:** 1Institute of Medical Research, Northwestern Polytechnical University, Xi’an 710072, China; xhwang616@mail.nwpu.edu.cn (X.W.); huhuijing@nwpu.edu.cn (H.H.); 2Department of Rehabilitation Medicine, The First Affiliated Hospital, Sun Yat-sen University, Guangzhou 510080, China; gele3@mail.sysu.edu.cn

**Keywords:** non-invasive brain stimulation, stroke, spasticity, meta-analysis

## Abstract

In recent years, the potential of non-invasive brain stimulation (NIBS) for the therapeutic effect of post-stroke spasticity has been explored. There are various NIBS methods depending on the stimulation modality, site and parameters. The purpose of this study is to evaluate the efficacy of NIBS on spasticity in patients after stroke. This systematic review and meta-analysis was conducted according to Preferred Reporting Items for Systematic reviews and Meta-Analyses (PRISMA) guidelines. PUBMED (MEDLINE), Web of Science, Cochrane Library and Excerpta Medica Database (EMBASE) were searched for all randomized controlled trials (RCTs) published before December 2021. Two independent researchers screened relevant articles and extracted data. This meta-analysis included 14 articles, and all included articles included 18 RCT datasets. The results showed that repetitive transcranial magnetic stimulation (rTMS) (MD = −0.40, [95% CI]: −0.56 to −0.25, *p* < 0.01) had a significant effect on improving spasticity, in which low-frequency rTMS (LF-rTMS) (MD = −0.51, [95% CI]: −0.78 to −0.24, *p* < 0.01) and stimulation of the unaffected hemisphere (MD = −0.58, [95% CI]: −0.80 to −0.36, *p* < 0.01) were beneficial on Modified Ashworth Scale (MAS) in patients with post-stroke spasticity. Transcranial direct current stimulation (tDCS) (MD = −0.65, [95% CI]: −1.07 to −0.22, *p* < 0.01) also had a significant impact on post-stroke rehabilitation, with anodal stimulation (MD = −0.74, [95% CI]: −1.35 to −0.13, *p* < 0.05) being more effective in improving spasticity in patients. This meta-analysis revealed moderate evidence that NIBS reduces spasticity after stroke and may promote recovery in stroke survivors. Future studies investigating the mechanisms of NIBS in addressing spasticity are warranted to further support the clinical application of NIBS in post-stroke spasticity.

## 1. Introduction

Post-stroke spasticity, as a neurological manifestation with a typical syndrome of increased muscle tone, was reported to have a prevalence rate of up to 25% in stroke survivors [1]. Spasticity leads to complications such as pain, muscle spasticity, abnormal joint positions and anchylosis, which further decrease the motor function of patients after stroke and bring great challenges to their daily activities [2]. Therefore, effective interventions for post-stroke spasticity are very important. Current management regimens for post-stroke spasticity include electrical stimulation of muscles, botulinum toxin injections, oral anti-spasticity drugs and wearable exoskeletons devices, etc. [3,4]. However, common side effects of drugs and the invasiveness of local treatment are undesirable, which limits their effectiveness.

In recent years, non-invasive brain stimulation (NIBS) has been actively explored in various diseases of the nervous system. Among various NIBS techniques, transcranial magnetic stimulation (TMS) and transcranial direct current stimulation (tDCS) are most often used to treat patients with post-stroke spasticity [5,6]. Spasticity usually occurs within one to six weeks after stroke and is caused by abnormal or hyperexcitable spinal reflexes [7,8]. NIBS induces excitatory changes in the underlying cerebral cortex in a non-invasive manner and lasting changes in neuroplasticity [9]. NIBS works by altering the excitability of the cerebral motor cortex and indirectly reducing the excitability of motor neurons in the spinal cord through the H-reflex [10].

Currently, the effects of NIBS on post-stroke spasticity are contradictory. Although some studies have reported a beneficial effect of NIBS in the treatment of post-stroke spasticity [11,12,13], other studies have shown no significant benefit of NIBS in reducing muscle spasticity. A meta-analysis published in 2020 showed no significant effect of rTMS in spasticity management. However, it included only five RCTs [14]. Results from two published meta-analyses of tDCS for post-stroke spasticity also showed some variability without uniform criteria [15,16]. Therefore, the aim of this study is to conduct a systematic review and meta-analysis of the effectiveness of NIBS in the management of spasticity in patients after stroke.

## 2. Methods

### 2.1. Literature Search Strategy

This meta-analysis was performed in accordance with the PRISMA guidelines for systematic reviews and meta-analysis [17]. The PICO principles consist of four parts: population, interventions, control and outcome and all articles included in systematic reviews and meta-analyses are retrieved according to the PICO principles [18]. The inclusion criteria for articles are (1) Population: patients who have been diagnosed as stroke patients by clinical examinations and have post-stroke spasticity; (2) Interventions: NIBS; (3) Control: sham stimulation; (4) Outcome: MAS; and (5) Research type: RCT. The research language is limited to English. Two authors independently searched electronic databases, including PUBMED (MEDLINE), Web of Science, Cochrane Library and EMBASE. We searched the database for related articles published as of December 2021 by using MeSH terms including “Stroke”, “Non-invasive Brain Stimulation” and “spasticity”. If there is a disagreement in the article inclusion process, it will be discussed with the third author to determine the eligibility for inclusion.

### 2.2. Study Selection

The article search strategy is shown in Figure 1. We retrieved a total of 2482 publications in our first search. The two authors screened titles and abstracts to determine relevant research articles and then further reviewed the full text to finally determine the research articles included in the meta-analysis. Any disagreements during the inclusion process were discussed and resolved by the third author.

### 2.3. Quality Assessment

All included RCTs were independently evaluated by two authors using the Cochrane risk of bias assessment tool [19]. It included six items: selection bias: random sequence generation and allocation concealment; performance bias: blinding of participants and personnel; detection bias: blinding of outcome assessment; attrition bias: incomplete outcome data; reporting bias: selective reporting; and other biases [20]. If there was a disagreement in the evaluation, it would be resolved through a discussion with the third author.

### 2.4. Data Extraction

For each study that met the inclusion criteria, relevant information about experimental design and result analysis was extracted. All extracted information included research characteristics (author, publication year and sample size), treatment parameters (stimulation method, stimulation parameters, stimulation time and control group) and main measurement results (MAS).

### 2.5. Statistical Analysis

A meta-analysis of the extracted studies was performed. Meta-analyses are useful for assessing the strength of evidence for treatment from multiple studies. The aim is to determine whether there is an effect, either positive or negative, and to obtain a single pooled estimate of effect rather than a single estimate of individual studies. In this meta-analysis, for each outcome related to continuous data, we calculated a pooled estimate and 95% confidence interval (CI) of the mean difference (MD) between the experimental and control groups after the intervention.

This meta-analysis used RevMan 5.4 (The Nordic Cochrane Centre, The Cochrane Collaboration, Copenhagen, Denmark) for statistical analysis. This was performed by entering the mean and standard deviation of all continuous data in each study into the software and calculating the mean difference (MD) of the 95% confidence interval (CI) to analyze the results. Cochran’s Q test and the I^2^ index were used to assess the heterogeneity of all studies included in the meta-analysis. Statistical heterogeneity between these studies was calculated using Cochran’s Q test and the I^2^ index. An I^2^ index > 50% and *p* < 0.10 of the Cochran’s Q test indicated high heterogeneity, and the random-effects model was used; otherwise, the fixed-effects model was used. The results of all data analyses in this meta-analysis were shown by forest plots.

Funnel plots and Egger’s test to assess potential publication bias were applied. Still, because the number of studies included in each meta-analysis was less than 10, the funnel plot and Egger’s test could produce misleading results in this case [21]. Therefore, the funnel plot and Egger’s test were not used in this meta-analysis to assess publication bias.

## 3. Results

### 3.1. Study Identification and Selection

A total of 2482 publications were retrieved from two authors independently by searching the database. The search results are shown in Figure 1. Of these, 1673 duplicate publications were firstly deleted, then 489 publications were screened based on titles, and then 287 publications were based on abstracts, and finally, 33 full-text articles were retrieved. Through the final full-text review, 14 articles were ultimately included for this review. This study included eight research articles [22,23,24,25,26,27,28,29] on rTMS, one of which included three data sets, one article included two data sets and the other articles each had one data set. A total of 128 patients received rTMS in all studies, and 104 patients served as the control group. At the same time, this study included six research articles [13,30,31,32,33,34] on tDCS. One article included two data sets, and the other articles had one data set. A total of 199 patients in all studies received tDCS, and 146 patients served as the control group. The information extracted from all research related to rTMS is shown in Table 1, and the information extracted from all studies related to tDCS is shown in Table 2.

Details of each study are provided in Table 1 and Table 2. In rTMS, the pooled sample size was 135 individuals receiving rTMS, with sample sizes ranging from 7 to 22 participants per group. In terms of study design, all articles in this review were RCTs. In tDCS, the pooled sample size was 196 individuals receiving tDCS, with sample sizes ranging from 10 to 45 participants per group. In terms of study design, all articles in this review were RCTs.

### 3.2. Effects of rTMS

A total of 11 RCTs on the effect of rTMS on post-stroke spasticity were included in the study, and the outcome measure of all the studies was MAS. The meta-analysis showed that compared with the control group, rTMS had significant benefits for patients with post-stroke spasticity, and the MAS was significantly reduced (MD: −0.40, 95% CI: −0.56 to −0.25, *p* < 0.01). The meta-analysis showed that there was no significant heterogeneity between the various studies (*p* = 0.42, I^2^ = 3%) (Figure 2A).

The different stimulation methods of rTMS were divided into different subgroups. Six of all studies used LF-rTMS, two studies used intermittent theta-burst rTMS (iTBS), and high-frequency rTMS (HF-rTMS), LF-rTMS combined with HF-rTMS and continuous theta-burst rTMS (cTBS) each had one study. The meta-analysis showed that compared with the control group, LF-rTMS had significant benefits for post-stroke spasticity, and the MAS was significantly reduced (MD: −0.51, 95% CI: −0.78 to −0.24, *p* < 0.01). However, although other studies had shown certain benefits, they did not reach statistical differences (Figure 2B).

The different stimulation sites of rTMS were divided into different subgroups. Six of the studies included the unaffected hemispheres of patients with post-stroke spasticity, and the other four studies included the affected hemispheres of patients. The meta-analysis showed that compared with the control group, rTMS applied to stimulate the unaffected hemispheres of patients with post-stroke spasticity had significant benefits, and the MAS was significantly reduced (MD: −0.58, 95% CI: −0.80 to −0.36, *p* < 0.01). However, stimulation of the affected hemispheres also had certain benefits but did not reach statistical differences (Figure 2C).

### 3.3. Effects of tDCS

A total of seven RCTs on the effects of tDCS on post-stroke spasticity were included in the study, and the measurement outcome for all studies was the MAS. The meta-analysis showed that compared with the control group, tDCS had significant benefits for patients with post-stroke spasticity, and the MAS was significantly reduced (MD: −0.65, 95% CI: −1.07 to −0.22, *p* < 0.01). This meta-analysis showed that there was heterogeneity between different studies (*p* < 0.01, I^2^ = 78%) (Figure 3A).

The stimulation types of tDCS were divided into different subgroups. Four studies used anodal stimulation, and three studies used cathodal stimulation. The meta-analysis showed that compared with the control group, anodal stimulation had significant benefits for patients with post-stroke spasticity (MD: −0.74, 95% CI: −1.35 to −0.13, *p* < 0.05); however, although cathode stimulation also had certain benefits, it did not reach a statistical difference (MD: −0.51, 95% CI: −1.31 to 0.29, *p* = 0.22) (Figure 3B).

The stimulation intensities of tDCS were divided into different subgroups. There were five studies with a stimulation intensity of 2.0 mA and the other two studies with a stimulation intensity of 0.7 mA and 1.2 mA, respectively. The meta-analysis showed that compared with the control group, the stimulation intensity of tDCS of 0.7 mA (MD: −1.20, 95% CI: −1.40 to −1.00, *p* < 0.01) and 1.2 mA (MD: −1.00, 95% CI: −1.26 to −0.74, *p* < 0.01) had significant effect on patients with post-stroke spasticity. However, the measurement results of other studies had changed but did not reach statistical differences (Figure 3C).

### 3.4. Risk of Bias and Sensitivity Analysis

In this meta-analysis, three of the included articles [26,28,31] designed different experimental groups based on the stimulation method. There was no mutual interference between the different experimental groups, so each study was treated as an RCT. Finally, a total of 18 studies were obtained from 14 articles in the meta-analysis. Two authors independently assessed the risk of bias assessment of 18 included studies. The results of the risk of bias for all studies are shown in Figure 4. The risk of bias was assessed using the Cochrane Collaboration recommendations, and the sensitivity results indicated that the results of our meta-analysis appeared to be stable [20].

## 4. Discussion

In this current study, a meta-analysis of the effect of NIBS on spasticity for post-stroke populations was performed. It included 18 RCTs, with the most relevant RCTs to date based on stringent inclusion and exclusion criteria. The results of the meta-analysis proved that NIBS has a positive effect on post-stroke spasticity. In addition, the sub-group analysis of NIBS (i.e., tDCS and TMS) on post-stroke spasticity was also conducted.

In terms of rTMS, the results of different subgroup analyses showed that LF-rTMS had a significant benefit in the unaffected hemispheres of patients with post-stroke spasticity (Figure 2B,C). This finding is in line with clinical evidence-based guidelines, which have shown that LF-rTMS acts on the unaffected hemisphere to promote post-stroke motor function recovery [35]. rTMS uses magnetic signals of different frequencies to stimulate the central nervous system in the corresponding parts and relieve limb spasticity in patients after stroke, and induce brain plasticity and brain network reorganization, promote the rehabilitation of the primary and secondary motor cortex [36]. Studies have shown that joint application of LF-rTMS acting on the unaffected hemisphere and HF-rTMS acting on the affected hemisphere can achieve better therapeutic effects by regulating the excitability of bilateral hemispheres [37]. However, there is no consistent standard for different stimulation methods. The possible mechanism of LF-rTMS for addressing spasticity may be related to the changes in the excitability of the cerebral motor cortex, thereby reducing the excitability of spinal motor neurons [13].

For the stimulation types of tDCS, anodal stimulation has significant benefits for spasticity treatment in post-stroke patients (Figure 3B). In terms of the stimulation strength, tDCS at current strengths of 0.7 mA or 1.2 mA significantly reduced spasticity, but the current strength of 2.0 mA showed no significant effect on post-stroke spasticity (Figure 3C). tDCS uses a low-intensity current to act on the target brain area to change the charge distribution of neuron membrane potential, resulting in depolarization or hyperpolarization, thereby changing the excitability of the cerebral cortex [38]. The anodal of tDCS is placed on the affected side to increase the excitability of the target brain area, and the cathodic is placed on the unaffected side to suppress the excitability of the target brain area. Studies have shown that anodal stimulation on the affected side can reduce limb spasticity symptoms in stroke survivors more than cathodal stimulation on the unaffected side [39]. The results of this meta-analysis are consistent with previous studies, which also showed a better effect of anodal tDCS on post-stroke spasticity. However, the mechanism of action of tDCS on post-stroke rehabilitation remains to be further investigated.

In the studies included in this meta-analysis, most of the brain regions stimulated by NIBS were the primary motor cortex [22,27,28,30], and a few studies were stimulated in the premotor cortex [23,30] and cerebellum [40]. The premotor cortex plays an important role in motor control and is another stimulation target besides the primary motor cortex [41,42]. The cerebellum works in concert with the cerebral cortex, is involved in motor control and has a role in the regulation of muscle tone [43]. The cerebellum may become a new target for NIBS in future studies. Although NIBS on different brain regions has rehabilitation effects on post-stroke spasticity, the interaction mechanism between different targets is still unclear. The mechanism of action between different targets needs to be further investigated in future studies.

There are several different scales for assessing spasticity in post-stroke patients in rehabilitation studies. Currently, the MAS is used in most studies, and its main purpose is to evaluate abnormal muscle tone, while a small number of studies use the Modified Tardieu Scale (MTS) as a spasticity assessment tool [44]. As the number of other scale studies (i.e., MTS) was too small, all studies included in this meta-analysis used MAS. However, both the MAS and MTS are subject to a certain degree of subjectivity, and more objective assessment methods need to be used in future research [45].

In patients after stroke, the balance between the two hemispheres of the brain is disrupted, resulting in hyperexcitability of the unaffected hemisphere and increased inhibition of the affected hemisphere [46]. Most of the reported findings showed that LF-rTMS had a positive effect on post-stroke spasticity [47,48,49]. Li et al. [50] showed that cTBS of the cerebellum reduced symptoms in patients with post-stroke spasticity. In addition, concomitant use of LF-rTMS and cTBS in post-stroke spastic patients resulted in better outcomes in rehabilitation. Different research results showed that different stimulation types of tDCS had certain therapeutic effects on patients with post-stroke spasticity [51,52,53,54]. The results of this meta-analysis are consistent with those of previous studies. Overall, NIBS for post-stroke spasticity is still mainly focused on the research of rTMS and tDCS, and the causal mechanisms underlying NIBS remain elusive. More comprehensive research is needed in the future.

Based on this meta-analysis, the results of non-randomized controlled trials of NIBS for post-stroke spasticity were also discussed. At this stage, no other NIBS have been found in RCTs of patients with post-stroke spasticity, and new techniques still need to be explored in future studies.

## 5. Conclusions

The results of the current meta-analysis are encouraging as they suggest that NIBS can promote rehabilitation in patients with post-stroke spasticity. At present, the NIBS applied to the field of post-stroke spasticity rehabilitation are mainly rTMS and tDCS. Other techniques, including transcranial alternating current stimulation (tACS) and transcranial ultrasound stimulation (TUS), still have limited evidence of significant variability in stimulation targets and stimulation parameters. Therefore, further in-depth study on the mechanism of action in the rehabilitation of post-stroke spastic patients is required. We hope that in the future, NIBS can be optimized and applied safely and efficiently to the rehabilitation of post-stroke spasticity.

## Figures and Tables

**Figure 1 brainsci-12-00836-f001:**
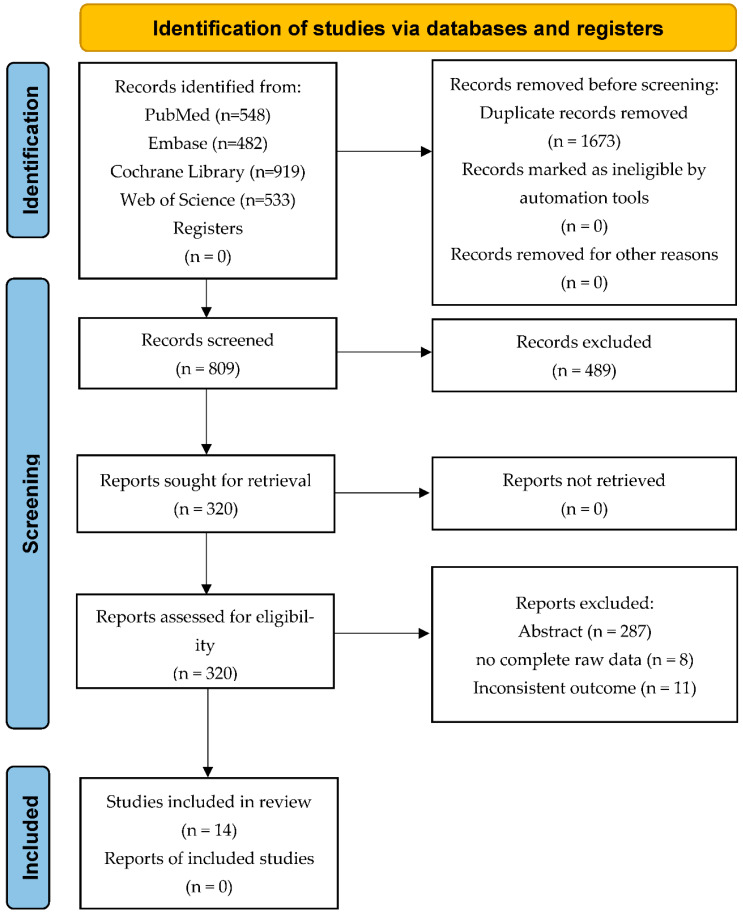
PRISMA flow diagram for search strategy and study selection.

**Figure 2 brainsci-12-00836-f002:**
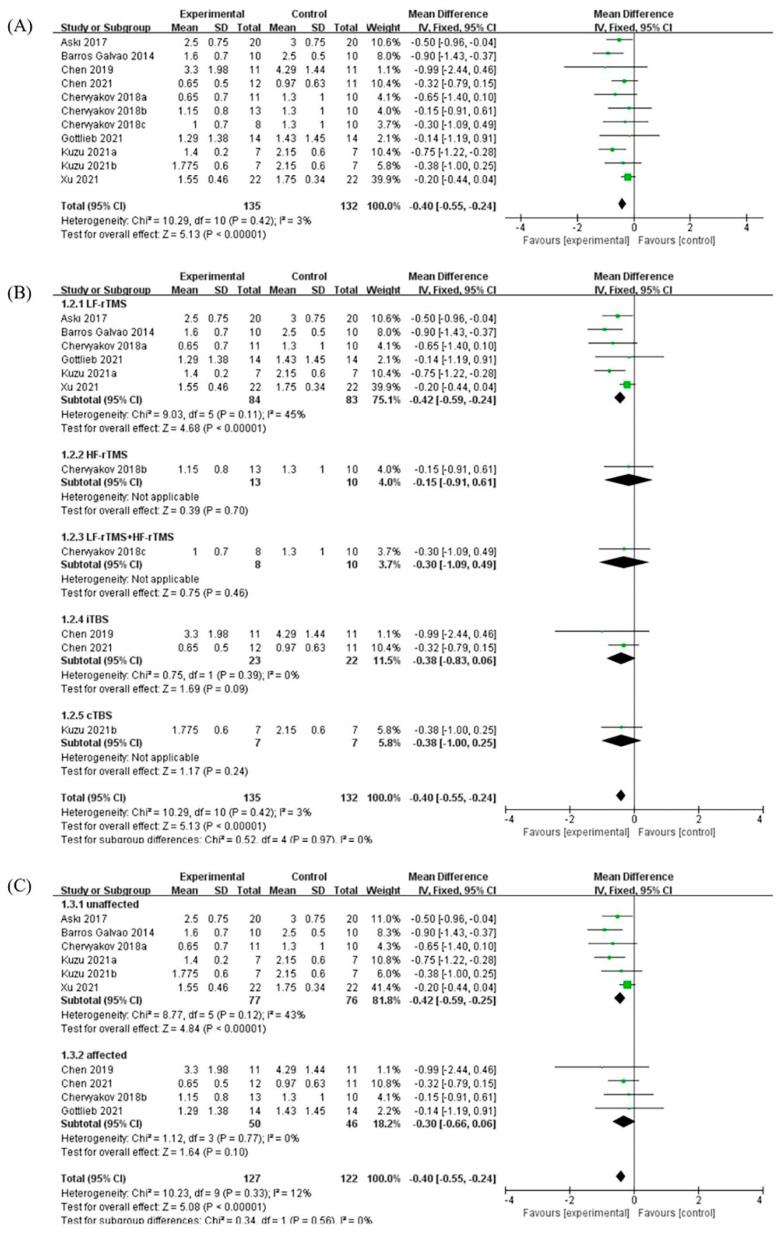
(**A**) Forest plot analysis of the effect of rTMS on post-stroke spasticity. (**B**) Forest plot analysis of the effects of different stimulation methods for rTMS on post-stroke spasticity. (**C**) Forest plot analysis of the effects of different stimulation sites for TMS on post-stroke spasticity.

**Figure 3 brainsci-12-00836-f003:**
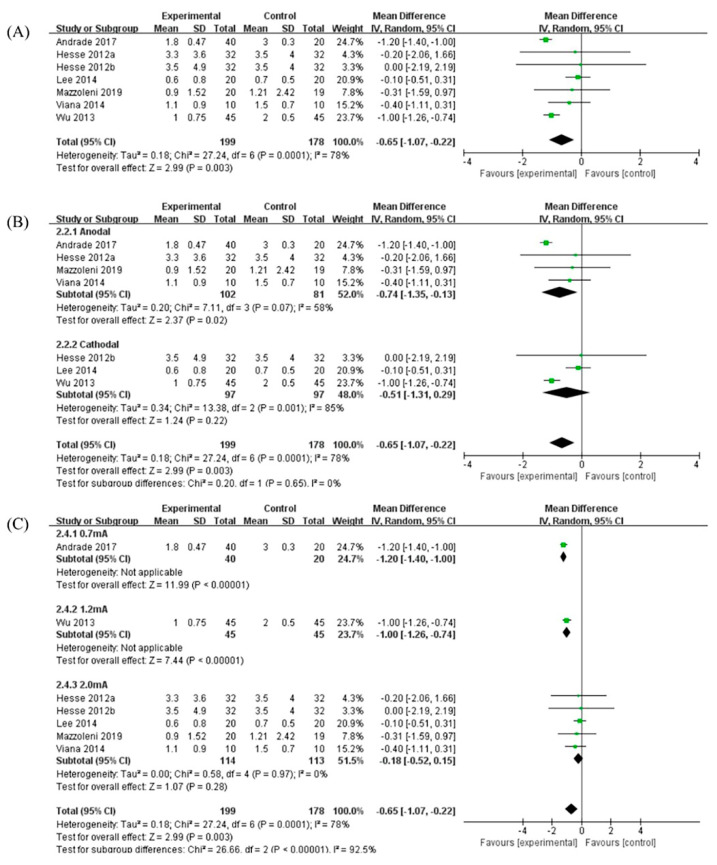
(**A**) Forest plot analysis of the effect of tDCS on post-stroke spasticity. (**B**) Forest plot analysis of the effects of different stimulation types for tDCS on post-stroke spasticity. (**C**) Forest plot analysis of the effects of different stimulation intensities for tDCS on post-stroke spasticity.

**Figure 4 brainsci-12-00836-f004:**
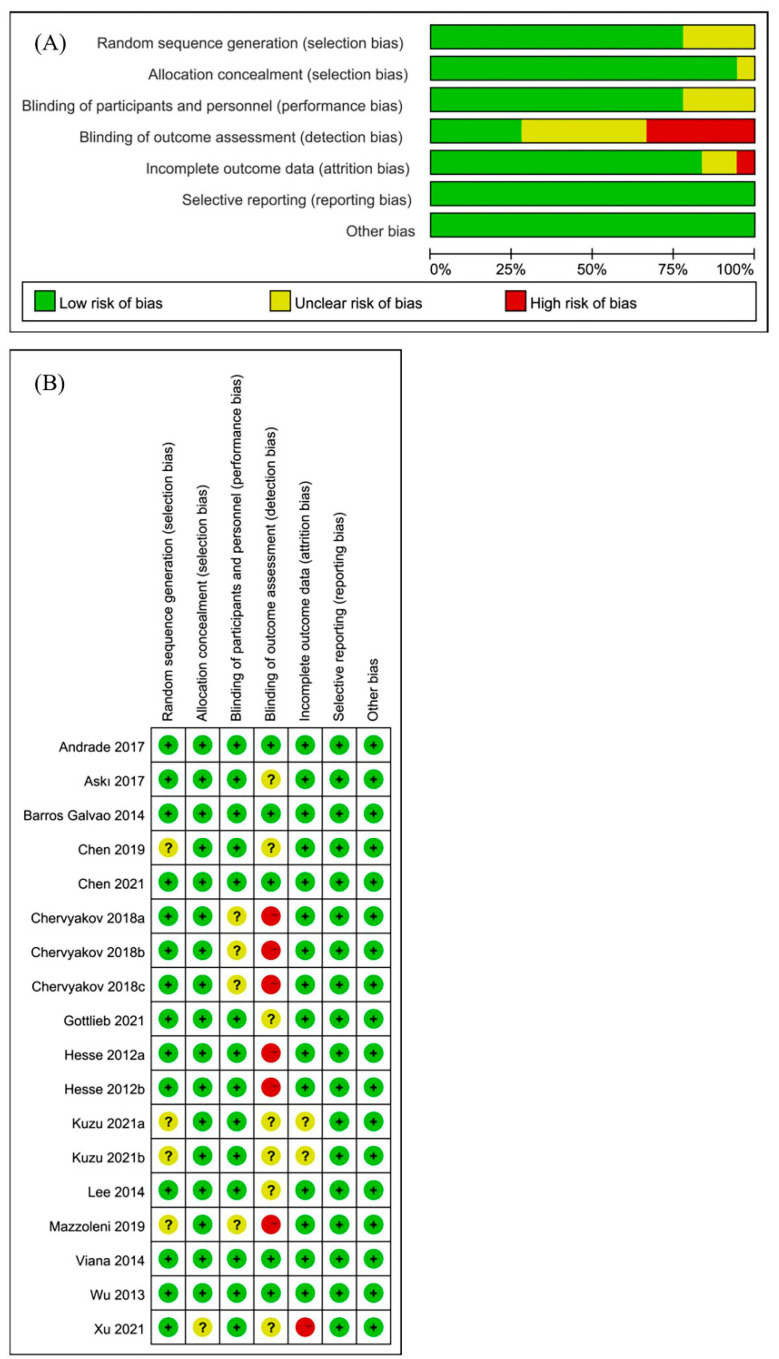
Risk of bias in the systematic review. (**A**) Risk of bias graph: review of the authors’ judgments about each risk of bias item, presented as percentages across all included studies. (**B**) Risk of bias summary: review of authors’ judgments about each risk of bias item for each included study.

**Table 1 brainsci-12-00836-t001:** Research Characteristics of rTMS.

Study	Participant				Mean Severity (SD)		Intervention	Control	Outcomes	Muscle
	Intervention		Control		Intervention	Control				
	Mean Age (SD)	N (Male/Female)	Mean Age (SD)	N (Male/Female)	MAS					
Askı et al. (2017)	56.75 (11.46)	20 (14/6)	58.80 (12.02)	20 (15/5)	3.2 (0.75)	2.8 (0.75)	LF-rTMS + PT 1200 pulses, 1 Hz, 90% RMT	Sham rTMS + PT	MAS	upper limb
Barros Galvao et al. (2014)	57.4 (12.0)	10 (6/4)	64.6 (6.8)	10 (7/3)	2.5 (0.5)	2.4 (0.5)	LF-rTMS + PT 1500 pulses, 1 Hz; 90% RMT	Sham rTMS +PT	MAS	wrist
Chen et al. (2019)	52.9 (11.1)	11 (7/4)	52.6 (8.3)	11 (7/4)	3.90 (2.10)	4.05 (1.56)	iTBS 50 Hz 80% AMT	Sham iTBS	MAS	upper limb
Chen et al. (2021)	54.36 (10.56)	12 (8/4)	48.95 (9.63)	11 (10/1)	0.87 (0.54)	0.94 (0.69)	iTBS + VCT 50 Hz 80% AMT	Sham iTBS + VCT	MAS	upper limb
Chervyakov et al. (2018a)	54.2 (11.1)	11 (5/6)	61.4 (11.4)	10 (5/5)	1.2 (0.9)	1.4 (1.0)	LF-rTMS 1200 pulses, 1 Hz, 100% RMT	Sham rTMS	MAS	arm
Chervyakov et al. (2018b)	58.6 (10.4)	13 (10/3)	61.4 (11.4)	10 (5/5)	1.84 (0.8)	1.4 (1.0)	HF-rTMS 200 pulses, 10-Hz, 80% RMT	Sham rTMS	MAS	arm
Chervyakov et al. (2018c)	60.7 (9.6)	8 (6/2)	61.4 (11.4)	10 (5/5)	1.5 (0.9)	1.4 (1.0)	LF-rTMS 1 Hz 100% RMT HF-rTMS 10 Hz 80% RMT	Sham rTMS	MAS	arm
Gottlieb et al. (2021)^]^	63.93 (10.91)	14 (9/5)	62.43 (11.46)	14 (3/11)	1.86 (1.35)	1.71 (1.27)	LF-rTMS 1200 pulses, 1 Hz	Sham-rTMS	MAS	upper limb
Kuzu et al. (2021a)	56.3 (11.5)	7 (4/3)	65.0 (4.6)	6 (2/4)	1.8 (0.4)	2.3 (0.6)	LF-rTMS 1200 pulses, 1 Hz	Sham rTMS	MAS	upper limb
Kuzu et al. (2021b)	61.3 (9.8)	7 (6/1)	65.0 (4.6)	6 (2/4)	2.1 (0.6)	2.3 (0.6)	cTBS 50 Hz	Sham cTBS	MAS	upper limb
Xu et al. (2021)	79.50 (1.49)	22 (17/5)	68.86 (3.09)	22 (15/7)	2.32 (0.48)	2.41 (0.50)	LF-rTMS + CRT 550 pulses, 1 Hz 90% RMT	Sham rTMS + CRT	MAS	upper limb

HF-rTMS: high-frequency repetitive transcranial magnetic stimulation; LF-rTMS: low-frequency repetitive transcranial magnetic stimulation; cTBS: continuous theta-burst repetitive transcranial magnetic stimulation; iTBS: intermittent theta-burst repetitive transcranial magnetic stimulation; AMT: active motor threshold; RMT: resting motor threshold; MAS: modified Ashworth scale; PT: physical therapy; VCT: virtual reality-based cycling training; CRT: conventional rehabilitation treatment.

**Table 2 brainsci-12-00836-t002:** Research Characteristics of tDCS.

Study	Participant				Mean Severity (SD)		Intervention	Control	Outcomes	Muscle
	Intervention		Control		Intervention	Control				
	Mean Age (SD)	N (Male/Female)	Mean Age (SD)	N (Male/Female)	MAS					
Andrade et al. (2017)	54.08 (3.72)	40 (22/18)	54.76 (4.28)	20 (12/8)	3.3 (0.36)	3.6 (0.5)	tDCS (Anodal) + CIMT 0.7 mA, 10 sessions	Sham-tDCS + CIMT	MAS	upper limb
Hesse et al. (2012a)	63.9 (10.5)	32 (20/12)	65.6 (10.3)	32 (21/11)	1.6 (2.9)	1.4 (2.7)	tDCS (Anodal) 2.0 mA, 30 sessions	Sham-tDCS	MAS	upper limb
Hesse et al. (2012b)	65.4 (8.6)	32 (18/14)	65.6 (10.3)	32 (21/11)	1.0 (1.8)	1.4 (2.7)	tDCS (Cathodal) 2.0 mA, 30 sessions	Sham-tDCS	MAS	upper limb
Lee and Chun (2014)	63.1 (10.3)	20 (12/8)	60.6 (14.1)	20 (9/11)	0.4 (0.5)	0.5 (0.4)	tDCS (Cathodal) + VRT 2.0 mA, 15 sessions	Sham-tDCS + VRT	MAS	upper limb
Mazzoleni et al. (2019)	67.50 (16.30)	20 (8/12)	68.74 (15.83)	19 (7/12)	1.1 (1.86)	1.58 (2.34)	tDCS (Anodal) + wrist robot-assisted rehabilitation 2.0 mA, 30 sessions	Sham-tDCS + wrist robot-assisted rehabilitation	MAS	wrist
Viana et al. (2014)	56.0 (10.2)	10 (9/1)	55.0 (12.2)	10 (7/3)	1.5 (0.7)	1.5 (0.52)	tDCS (Anodal) + VRT 2.0 mA, 15 sessions	Sham-tDCS + VRT	MAS	upper limb
Wu et al. (2013)	45.9 (11.2)	45 (34/11)	49.3 (12.6)	45 (35/10)	2.0 (0.75)	2.0 (0.5)	tDCS (Cathodal) + PT 1.2 mA, 20 sessions	Sham-tDCS + PT	MAS	elbow, wrist

tDCS: transcranial direct current stimulation; MAS: modified Ashworth scale; CIMT: constraint-induced movement therapy; VRT: virtual reality therapy; PT: physical therapy.

## Data Availability

Not applicable.

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
