# Peer review of "Effects of Non-Invasive Brain Stimulation on Post-Stroke Spasticity: A Systematic Review and Meta-Analysis of Randomized Controlled Trials"

_brainsci, 2022, doi:10.3390/brainsci12070836_

Round 1

Reviewer 1 Report

Abstract:

·       Minor change: include p-values for rTMS results.

Introduction:

·       Minor change: “Therefore, effective interventions for post-stroke spasticity are very important. Current management regimens for post-stroke spasticity include electrical stimulation of muscles, botulinum toxin injections, oral anti-spasticity drugs, etc. However, common side effects of drugs and the invasiveness of local treatment are undesirable which limits their effectiveness”. Please, include reference.

·        Minor change: NIBS mainly includes transcranial magnetic stim-ulation (TMS), transcranial electrical stimulation (tES) and transcranial ultrasound stimu-lation (TUS).  Modify this sentence, as NIBS does not "mainly include" these techniques: these are examples of stimulation, but them need not be the main techniques, as there are many more, such as tDCS, which is more studied than TUS for example.

·       Major change The association between spasticity and how NIBS might reduce it is unclear. Includes sentence: “and reducing the excitability of motor neurons in the spinal cord”, but it is not something that happens directly, as it happens through cortical modulation. As a suggestion, include H-reflex data.

·       Major change: I would like to see more justification of the study, or at least include similar articles and their differences, since the conclusions are different with respect to rTMS: Xu, P., Huang, Y., Wang, J. et al. Repetitive transcranial magnetic stimulation as an alternative therapy for stroke with spasticity: a systematic review and meta-analysis. J Neurol 268, 4013–4022 (2021). https://doi.org/10.1007/s00415-020-10058-4

There is also a lot of information on tDCS; what's new about it?:

Alashram, A. R., Padua, E., Aburub, A., Raju, M., & Annino, G. (2022). Transcranial direct current stimulation for upper extremity spasticity rehabilitation in stroke survivors: A systematic review of randomized controlled trials. PM&R.

Huang, J., Qu, Y., Liu, L., Zhao, K., & Zhao, Z. (2021). Efficacy and safety of transcranial direct current stimulation for post-stroke spasticity: A meta-analysis of randomised controlled trials. Clinical Rehabilitation, 02692155211038097.

Methods:

·       Included are studies comparing NIBS with placebo, and others that compared with traditional physical therapy. Perhaps they should be analyzed separately.

·       The article search specifies only NIBS, which is too broad a term, since only TMS and tDCS are studied.

·       Major change. It is unclear whether spasticity in the upper limb, lower Limb, or both is analyzed. It can be included in the tables.

·       Minor change: The number of studies obtained in the search should be included in the results, not in the study selection strategy.

·       Major change: Include data on the interpretation of Cochran's Q test (e.g., with P values < 0.05 considered significant), and the I2 index. Why was a random-effects model chosen and not a fixed model, even though the heterogeneity among some of the studies is 0% according to the I2 index?

·       Minor change: Indicate that the data were reflected in forest plots.

·       Major change: why were funnel plots not included to assess publication bias?

·       Major change: Reflect how they are going to analyze the results and do the meta-analysis, including whether they analyze techniques together, separately, or how.

Results:

·       Minor change: Perhaps exclude from the meta-analysis the techniques that only have 1-2 RCTs, and do the analysis separately for rTMS, and LF-rTMS.

·       Major change: In the tDCS results, the study by Andrade et al. and Wu et al. may be influential cases, as they contribute 24.7 and 23.7 of the weight, respectively. A new meta-analysis should be done without them and compare the results with and without these studies, and study the differences with the other included studies, as they contribute too much weight to the overall pooled result.

·       Same with the different types of tDCSCon intensidades, solo se debería analizar los estudios de 2.0 mA.

·       In the intensity section, only the 2.0 mA studies should be analyzed, since the rest of the intensities are provided by only one study.

Author Response

Abstract:

  1. 1. Minor change: include p-values for rTMS results.

Response to the comment: Thank you very much for pointing out the problems and the suggestion. We have modified it in the abstract section. It reads,

“The results showed that repetitive transcranial magnetic stimulation (rTMS) (MD = -0.40, [95% CI] : −0.56 to −0.25, p < 0.01) had a significant effect on improving spasticity, in which low-frequency rTMS (LF-rTMS) (MD = -0.51, [95% CI] : −0.78 to −0.24, p < 0.01) and stimulation of the unaffected hemisphere (MD = -0.58, [95% CI] : −0.80 to −0.36, p < 0.01) were beneficial on Modified Ashworth Scale (MAS) in patients with post-stroke spasticity.”

Changes in the manuscript: Page 1 Line 21-25.

Introduction:

  1. 2. Minor change: “Therefore, effective interventions for post-stroke spasticity are very important. Current management regimens for post-stroke spasticity include electrical stimulation of muscles, botulinum toxin injections, oral anti-spasticity drugs, etc. However, common side effects of drugs and the invasiveness of local treatment are undesirable which limits their effectiveness”. Please, include reference.

Response to the comment: Thank you very much for the positive comment. We have modified it in the introduction section. It reads,

“Current management regimens for post-stroke spasticity include electrical stimulation of muscles, botulinum toxin injections, oral anti-spasticity drugs and wearable exoskeletons devices etc [3, 4].”

Changes in the manuscript: Page 1 Line 40-42.

  1. Minor change: NIBS mainly includes transcranial magnetic stimulation (TMS), transcranial electrical stimulation (tES) and transcranial ultrasound stimulation (TUS). Modify this sentence, as NIBS does not "mainly include" these techniques: these are examples of stimulation, but them need not be the main techniques, as there are many more, such as tDCS, which is more studied than TUS for example.

Response to the comment: Thank you for the comment. As you suggested, we have modified it in the introduction section. It reads,

“Among various NIBS techniques, transcranial magnetic stimulation (TMS) and transcranial direct current stimulation (tDCS) are most often used to treat patients with post-stroke spasticity [5, 6].”

Changes in the manuscript: Page 2 Line 46–48.

  1. Major change: The association between spasticity and how NIBS might reduce it is unclear. Includes sentence: “and reducing the excitability of motor neurons in the spinal cord”, but it is not something that happens directly, as it happens through cortical modulation. As a suggestion, include H-reflex data.

Response to the comment: Thank you very much for the suggestion. We have modified it in the introduction section. It reads,

“NIBS works by altering the excitability of the cerebral motor cortex and indirectly reducing the excitability of motor neurons in the spinal cord through the H-reflex [10].”

Changes in the manuscript: Page 2 Line 51–53.

  1. Major change: I would like to see more justification of the study, or at least include similar articles and their differences, since the conclusions are different with respect to rTMS.

Response to the comment: Thank you very much for the positive comment. We have modified it in the introduction section. It reads,

“A meta-analysis published in 2020 showed no significant effect of rTMS in spasticity management. However it included only five RCTs [14]. Results from two published me-ta-analyses of tDCS for post-stroke spasticity also showed some variability without uniform criteria [15, 16].”

Changes in the manuscript: Page 2 Line 57-60.

Methods:

6.Included are studies comparing NIBS with placebo, and others that compared with traditional physical therapy. Perhaps they should be analyzed separately.

Response to the comment: Thank you very much for the suggestion. In our included studies, traditional physical therapy was used in both the control and experimental groups, and we compared the results of non-invasive brain stimulation techniques and sham stimulation. It reads,

“The inclusion criteria for articles are: (1) Population: patients who have been diagnosed as stroke patients by clinical examinations and have post-stroke spasticity; (2) Interventions: NIBS; (3) Control: sham stimulation; (4) Outcome: MAS; (5) Research type: RCT.”

Changes in the manuscript: Page 2 Line 68-71.

  1. The article search specifies only NIBS, which is too broad a term, since only TMS and tDCS are studied.

Response to the comment: Thank you for the comment. The aim of this study is to evaluate the efficacy of non-invasive brain stimulation (NIBS) on spasticity in patients after stroke. As a matter of fact, among various NIBS techniques, TMS and tDCS are most often used to treat patients with post-stroke spasticity. Following the literature search strategy, after study selection, our results showed that only TMS and tDCS met the research conditions. In this case, only TMS and tDCS were studied. In our discussion section, we also talked about it as follows:

“Based on this meta-analysis, we also focused on and discussed the results of non-randomized controlled trials of NIBS for post-stroke spasticity. At this stage, no other NIBS have been found in RCTs of patients with post-stroke spasticity, and new techniques still need to be explored in future studies.”

  1. Major change. It is unclear whether spasticity in the upper limb, lower Limb, or both is analyzed. It can be included in the tables.

Response to the comment: Thank you very much for suggestions. We have modified it in the results section. Please see the Table 1 and Table2.

Changes in the manuscript: Page 5-8 Table 1 and Table 2.

  1. Minor change: The number of studies obtained in the search should be included in the results, not in the study selection strategy.

Response to the comment: Thank you very much for the positive comment. We have modified it in the results section. It reads,

“A total of 2482 publications were retrieved from two authors independently by searching the database. The search results were shown in Figure 1. 1673 duplicate publications were firstly deleted, then 489 publications were screened based on titles, and then 287 publications were based on abstracts, and finally 33 full-text articles were retrieved. Through the final full-text review, 14 articles were ultimately included for this review.”

Changes in the manuscript: Page 4 Line 166–171.

  1. Major change: Include data on the interpretation of Cochran's Q test (e.g., with P values < 0.05 considered significant), and the I2 index. Why was a random-effects model chosen and not a fixed model, even though the heterogeneity among some of the studies is 0% according to the I2 index?

Response to the comment: Thank you for the comment. We have modified it in the methods section. An I2 index > 50 % and P < 0.05 of the Cochran's Q test indicated high heterogeneity, and the random-effects model was used; otherwise, the fixed-effects model was used. The results of all data analyses in this meta-analysis were shown by forest plots. Please see page line 155-157.

“An I2 index > 50% and P < 0.10 of the Cochran's Q test indicated high heterogeneity, and the random-effects model was used; otherwise, the fixed-effects model was used.”

Changes in the manuscript: Page 4 Line 155–157, Figure 2.

  1. Minor change: Indicate that the data were reflected in forest plots.

Response to the comment: Thank you very much for the suggestions. We have modified it in the methods section. It reads,

“The results of all data analyses in this meta-analysis were shown by forest plots.”

Changes in the manuscript: Page 4 Line 157-158.

  1. Major change: why were funnel plots not included to assess publication bias?

Response to the comment: Thank you for the comment. We have modified it in the methods section. In this meta-analysis, the sample size of both studies on transcranial magnetic stimulation (TMS) and transcranial direct current stimulation (tDCS) did not exceed 10, so we did not use funnel plots for publication bias assessment. It reads,

“Funnel plots and Egger's test to assess potential publication bias were applied, but because the number of studies included in each meta-analysis was less than 10, the funnel plot and Egger's test could produce misleading results in this case [21]. Therefore, funnel plot and Egger's test were not used in this meta-analysis to assess publication bi-as.”

Changes in the manuscript: Page 4 Line 159-163.

  1. Major change: Reflect how they are going to analyze the results and do the meta-analysis, including whether they analyze techniques together, separately, or how.

Response to the comment: Thank you for the comment. We have modified it. It reads,

“This meta-analysis used RevMan 5.4 (The Nordic Cochrane Centre, The Cochrane Collaboration) for statistical analysis. Enter the mean and standard deviation of all continuous data in each study into the software, and calculate the mean difference (MD) of the 95% confidence interval (CI) to analyze the results. Cochran’s Q test and the I2 index were used to assess the heterogeneity of all studies included in the meta-analysis. Statis-tical heterogeneity between these studies was calculated using Cochran's Q test and I2 index.”

Results:

  1. Minor change: Perhaps exclude from the meta-analysis the techniques that only have 1-2 RCTs, and do the analysis separately for rTMS, and LF-rTMS.

Response to the comment: Thank you for the advice. It is true that the number of 1-2 RCTs was a little small, but combined with other studies we had consulted, we found that the results showed a certain trend and significance, so we chose to retain this result.

The separate analysis results for rTMS and LF-rTMS are shown in the figure below. The results of rTMS analysis alone have been presented in the manuscript (Figure 2(A)). For the separate analysis results of LF-rTMS we can use as supplementary material if you think it is important.

  1. Major change: In the tDCS results, the study by Andrade et al. and Wu et al. may be influential cases, as they contribute 24.7 and 23.7 of the weight, respectively. A new meta-analysis should be done without them and compare the results with and without these studies, and study the differences with the other included studies, as they contribute too much weight to the overall pooled result.

Response to the comment: Thank you for the comment, which is very accurate and valuable. The two studies you mentioned do account for a large proportion of the overall study, but if we remove these two studies, other studies would have a large proportion in turn. After another meta-analysis, we believe that the results of studies that do not remove these two are more reliable.

  1. Same with the different types of tDCS Con intensidades, solo se debería analizar los estudios de 2.0 mA.
  2. In the intensity section, only the 2.0 mA studies should be analyzed, since the rest of the intensities are provided by only one study.

Response to the comment: Thank you very much for the positive comment. Your comments are correct and reasonable, we did not perform a meta-analysis of the results of only one study. Since there are few studies on the use of tDCS in the treatment of post-stroke spasticity, we hoped to include as many relevant studies as possible. There is currently no clear standard for the stimulation intensity of tDCS, so we hope to clarify the different intensities through discussion.

Reviewer 2 Report

In this manuscript, the authors conducted a systematic review and Meta-analysis to evaluate the effects of non-invasive brain stimulation (NIBS) on post-stroke induced spasticity. This study included 8 research articles on rTMS and 6 research articles on tDCS. Subgroup analysis was conducted based on the different stimulation methods, sites, and intensities. The results suggest that NIBS may promote recovery in patients with post-stroke induced spasticity. Furthermore, this review provided the evidence that the stimulation intensity, site, and methods are critical. Certain types of NIBS with certain intensities and sites could significantly improve the stroke-induced spasticity, whereas others may not. This information will be helpful for clinicians to develop an additional effective rehabilitation therapy plan for post-stroke spasticity patients.

There are a few minor suggestions:

1.       Page2 line 4-6: Need to add references here.

2.       Page 3 Fig.1.: Identification:  Need to report the number of records identified from each database rather than the total number across all databases.

3.       Page 3 Fig.1.: Reports sought for retrieval: (n=0). Should that be n=320?

4.       Page 3 2.5. Statistical analysis:  Need to classify the level of heterogeneity by I2 measurement. Need to explain why adopt random-effects model regardless of the difference in heterogeneity.

Author Response

In this manuscript, the authors conducted a systematic review and Meta-analysis to evaluate the effects of non-invasive brain stimulation (NIBS) on post-stroke induced spasticity. This study included 8 research articles on rTMS and 6 research articles on tDCS. Subgroup analysis was conducted based on the different stimulation methods, sites, and intensities. The results suggest that NIBS may promote recovery in patients with post-stroke induced spasticity. Furthermore, this review provided the evidence that the stimulation intensity, site, and methods are critical. Certain types of NIBS with certain intensities and sites could significantly improve the stroke-induced spasticity, whereas others may not. This information will be helpful for clinicians to develop an additional effective rehabilitation therapy plan for post-stroke spasticity patients.

Response: Thank you very much for the positive comments. The manuscript has been revised and the amendments were underlined in red.

There are a few minor suggestions:

1.Page 2 line 4 - 6: Need to add references here.

Response to the comment: Thank you very much for the positive comment. We have modified it in the introduction section. It reads,

“Among various NIBS techniques, transcranial magnetic stimulation (TMS) and transcranial direct current stimulation (tDCS) are most often used to treat patients with post-stroke spasticity [5, 6].”

Changes in the manuscript: Page 2 Line 46-48.

  1. Page 3 Fig.1.: Identification:  Need to report the number of records identified from each database rather than the total number across all databases.

Response to the comment: Thank you for the comment. We have modified it in the methods section. Please see the figure 1.

Changes in the manuscript: Page 3 Figure 1.

  1. Page 3 Fig.1.: Reports sought for retrieval: (n=0). Should that be n=320?

Response to the comment: Thank you very much for your comment. We have modified it in the methods section. Please see the figure 1.

Changes in the manuscript: Page 3 Figure 1.

  1. Page 3 2.5. Statistical analysis:  Need to classify the level of heterogeneity by I2 measurement. Need to explain why adopt random-effects model regardless of the difference in heterogeneity.

Response to the comment: Thank you very much for suggestions. We have modified it in the methods section. It reads,

“An I2 index > 50% and P < 0.10 of the Cochran's Q test indicated high heterogeneity, and the random-effects model was used; otherwise, the fixed-effects model was used.”

Changes in the manuscript: Page 4 Line 154–157, Figure 2.

Reviewer 3 Report

In this paper, the authors presented a review study of the evaluation of non-invasive brain stimulation on spasticity in patients after stroke. In total, 14 selected papers that include 18 randomized controlled trial datasets were presented and discussed. While the authors did a good survey and summary of the existing literature studies, there are still some issues and questions in this manuscript. The language and organization need to be further polished by a native speaker or technical writing assistant. More specific comments and concerns are listed in the PDF file.

Author Response

In this paper, the authors presented a review study of the evaluation of non-invasive brain stimulation on spasticity in patients after stroke. In total, 14 selected papers that include 18 randomized controlled trial datasets were presented and discussed. While the authors did a good survey and summary of the existing literature studies, there are still some issues and questions of this manuscript. The language and organization need to be further polished by a native speaker or technical writing assistant. More specific comments and concerns are listed below.

Response: Thank you very much for the positive comments. The manuscript has been revised and the amendments were underlined in red.

  1. As a review paper, the abstract is supposed to summarize the research topic, literature overview, main findings, challenges, possible future directions, and so on. However, too many specific results were included in the abstract section, mainly from three references. This needs to be significantly improved for a better presentation.

Response to the comment: Thank you very much for the positive comment. We have modified it in the abstract section. It reads,

“In recent years, the potential of non-invasive brain stimulation (NIBS) for the therapeutic effect of post-stroke spasticity has been explored. There are various NIBS methods depending on the stimulation modality, site and parameters. The purpose of this study is to evaluate the efficacy of NIBS on spasticity in patients after stroke.”

“This meta-analysis revealed moderate evidence that NIBS reduces spasticity after stroke and may promote recovery in stroke survivors.  Future studies of investigating the mechanisms of NIBS on addressing spasticity are warrant to further support the clinical application of NIBS in post-stroke spasticity.”

Changes in the manuscript: Page 1 Line 12–15, Line 28-31.

  1. In the abstract, some acronyms were directly used without full name definition, like PRISMA and EMBASE.

Response to the comment: Thank you very much for suggestions. We have modified it in the abstract section. It reads,

“This systematic review and meta-analysis was conducted according to Preferred Reporting Items for Systematic reviews and Meta-Analyses (PRISMA) guidelines. PUBMED (MEDLINE), Web of Science, Cochrane Library and Excerpta Medica Database (EMBASE) were searched for all randomized controlled trials (RCTs) published before December 2021.”

Changes in the manuscript: Page 1 Line 15–19.

  1. In the introduction, except for these mentioned management regimens, wearable exoskeleton rehabilitation devices are also an effective intervention for post-stroke spasticity, which could also attach more readers' interest.

Response to the comment: Thank you very much for the positive comment. We have modified it in the introduction section. It reads,

“Current management regimens for post-stroke spasticity include electrical stimulation of muscles, botulinum toxin injections, oral anti-spasticity drugs and wearable exoskeletons devices etc [3, 4].”

Changes in the manuscript: Page 1 Line 40–42.

  1. Once the acronym was defined, it was supposed to use the acronym throughout the sections in the paper, like NIBS. Please check all acronyms throughout the paper for a better consistency.

Response to the comment: Thank you very much for suggestions. We have checked and corrected all the problems you mentioned about abbreviations and their consistency.

Changes in the manuscript: Page 2 Line 51, Line 58, Page 14 Line 348.

  1. More specifications are needed for the PICO principles.

Response to the comment: Thank you very much for the positive comment. We have modified it in the methods section. It reads,

“The PICO principles consist of four parts: population, interventions, control and out-come and all articles included in systematic reviews and meta-analysis were retrieved according to the PICO principles [18].”

Changes in the manuscript: Page 2 Line 66–68.

  1. Contradictory statements were presented in the last sentence of section 2.1 and section 2.2.

Response to the comment: Thank you very much for suggestions. We have modified it in the methods section. It reads,

“If there is a disagreement in the article inclusion process, it will be discussed by the third author to determine the eligibility for inclusion.”

“Any disagreements during the inclusion process were discussed and resolved by the third author.”

Changes in the manuscript: Page 2 Line 76–77, Line 82-83.

  1. The fonts need to be increased and the resolution needs to be improved in Figure 1.

Response to the comment: Thank you very much for the positive comment. We have modified it in the methods section. Please see the figure 1.

Changes in the manuscript: Page 3 Figure 1.

  1. In section 2.3, the reference is required for the description of these six items.

Response to the comment: Thank you for the comment. We have modified it in the methods section. It reads,

“It included six items: selection bias: random sequence generation and allocation concealment; performance bias: blinding of participants and personnel; detection bias: blinding of outcome assessment; attrition bias: incomplete outcome data; reporting bias: selective reporting; and other biases [20].”

Changes in the manuscript: Page 3 Line 129-132.

  1. In section 2.5, it was not clear how the statistical analysis was performed. Please specify the purpose of this statistical analysis, continuous data in each study, and the meta-analysis.

Response to the comment: Thank you very much for suggestions. We have modified it in the methods section. It reads,

“A meta-analysis of the extracted studies was performed. Meta-analyses are useful for assessing the strength of evidence for treatment from multiple studies; the aim is to determine whether there is an effect, whether positive or negative, and to obtain a single pooled estimate of effect, rather than a single estimate of individual studies. In this me-ta-analysis, for each outcome related to continuous data, we calculated a pooled estimate and 95% confidence interval (CI) of the mean difference (MD) between the experimental and control groups after the intervention.

This meta-analysis used RevMan 5.4 (The Nordic Cochrane Centre, The Cochrane Collaboration) for statistical analysis. Enter the mean and standard deviation of all continuous data in each study into the software, and calculate the mean difference (MD) of the 95% confidence interval (CI) to analyze the results. Cochran’s Q test and the I2 index were used to assess the heterogeneity of all studies included in the meta-analysis. Statistical heterogeneity between these studies was calculated using Cochran's Q test and I2 index. An I2 index > 50% and P < 0.10 of the Cochran's Q test indicated high heterogeneity, and the random-effects model was used; otherwise, the fixed-effects model was used. The results of all data analyses in this meta-analysis were shown by forest plots.

Funnel plots and Egger's test to assess potential publication bias were applied, but because the number of studies included in each meta-analysis was less than 10, the funnel plot and Egger's test could produce misleading results in this case [21]. Therefore, funnel plot and Egger's test were not used in this meta-analysis to assess publication bi-as.”

Changes in the manuscript: Page 4 Line 142-163.

  1. In Table 1, there were no definitions of cTBS, iTBS, and AMT. Also, the information in the intervention box was hard to be differentiated from multiple reference studies. This table needs to be re-arranged to clearly present studies summary.

Response to the comment: Thank you very much for the positive comment. We have modified it in the results section. Please see the Table 1.

“cTBS: continuous theta-burst repetitive transcranial magnetic stimulation; iTBS: intermittent theta-burst repetitive transcranial magnetic stimulation; AMT: ac-tive motor threshold; RMT: resting motor threshold;”

Changes in the manuscript: Page 7 Line 187–189 and Table 1.

  1. Only the list of each reference study's characteristics was not enough, and further clarifications and discussions of Table 1 and Table 2 are needed.

Response to the comment: Thank you very much for the positive comment. We have modified it in the results section. It reads,

“Details of each study were provided in Tables 1 and Table 2. In rTMS, the pooled sample size was 135 individuals receiving rTMS, with sample sizes ranging from 7 to 22 participants per group. In terms of study design, all articles in this review were RCTs. In tDCS, the pooled sample size was 196 individuals receiving tDCS, with sample sizes ranging from 10 to 45 participants per group. In terms of study design, all articles in this review were RCTs.”

Changes in the manuscript: Page 4 Line 179–184.

  1. In section 3.2, results of significant MAS reduction were reported but no baseline information was provided for these comparisons.

Response to the comment: Thank you very much for the positive comment. We have modified it in the results section. Baseline data on MAS scores for all our experimental and control groups are shown in Table 1 and Table 2. The spasticity treatment effect of our experimental group was compared with that of the control group. It reads,

“The meta-analysis showed that compared with the control group, rTMS had significant benefits for patients with post-stroke spasticity, and the MAS was significantly reduced (MD: -0.40, 95% CI : −0.56 to −0.25, p < 0.01).”

“The meta-analysis showed that compared with the control group, LF-rTMS had signifi-cant benefits for post-stroke spasticity, and the MAS was significantly reduced (MD: -0.51, 95% CI : −0.78 to −0.24, p < 0.01).”

“The different stimulation sites of rTMS were divided into different subgroups. Eight of the studies included the unaffected hemispheres of patients with post-stroke spasticity, and the other three studies included the affected hemispheres of patients.”

Changes in the manuscript: Page 8 Line 210–213, Line 218-221, Line 223-225.

  1. In section 3.2.3, “eight of the studies” and “the other four studies” did not match the reference studies' numbers. The similar issues could also be found in section 3.2.6.

Response to the comment: Thank you very much for suggestions. We have modified it in the results section. We have tried our best to avoid the problems in the revision. It reads,

“Eight of the studies included the unaffected hemispheres of patients with post-stroke spasticity, and the other three studies included the affected hemispheres of patients.”

“There were five studies with stimulation intensity of 2.0 mA, and the other two studies with stimulation intensity of 0.7 mA and 1.2 mA respectively.”

Changes in the manuscript: Page 8 Line 224–225, Page 10 Line 249-251.

  1. From Fig. 2 to Fig. 7, many entries on the left side were repeated several times, which did not provide more useful information but took much space in the manuscript. The authors need to come up with a better way to summarize and report those results in section 3.2.

Response to the comment: Thank you very much for suggestions. We have modified it in the results section. We have merged figures 2-4 in the original manuscript into figures 2 in the revised manuscript and divided them into three parts A, B, and C. Meanwhile, we have merged figures 5-7 in the original manuscript into figure 3 in the revised manuscript, and divided them into three parts A, B, and C. Please see the figure 2 and figure 3.

Changes in the manuscript: Page 8 figure 2, Page 9 figure 3.

  1. Consider separating the current section 3.2 into section 3.2 and section 3.3, and each one has one type intervention. In the current organization, it was really hard to tell the connection or difference among section 3.2.1 and its subsequent sections 3.2.2 and 3.2.3, and among section 3.2.4 and its subsequent sections 3.2.5 and 3.2.6.

Response to the comment: Thank you very much for suggestions. We have modified it in the results section. We merged sections 3.2.1-3.2.3 in the original manuscript into section 3.2 in the revised manuscript. We merged sections 3.2.4-3.2.6 in the original manuscript into section 3.3 in the revised manuscript. It reads,

“A total of 11 RCTs on the effect of rTMS on post-stroke spasticity were included in the study, and the outcome measure of all the studies was MAS. The meta-analysis showed that compared with the control group, rTMS had significant benefits for patients with post-stroke spasticity, and the MAS was significantly reduced (MD: -0.40, 95% CI : −0.56 to −0.25, p < 0.01). The meta-analysis showed that there was no significant heterogeneity between the various studies (P = 0.42, I2 = 3 %) (Figure 2 (A)). A total of 11 RCTs on the effect of rTMS on post-stroke spasticity were included in the study, and the outcome measure of all the studies was MAS. The meta-analysis showed that compared with the control group, rTMS had significant benefits for patients with post-stroke spasticity, and the MAS was significantly reduced (MD: -0.40, 95% CI : −0.56 to −0.25, p < 0.01). The meta-analysis showed that there was no significant hetero-geneity between the various studies (P = 0.42, I2 = 3 %) (Figure 2 (A)).

The different stimulation methods of rTMS were divided into different subgroups. Six of all studies used LF-rTMS, two studies used intermittent theta-burst rTMS (iTBS), and high-frequency rTMS (HF-rTMS), LF-rTMS combined with HF-rTMS and continuous theta-burst rTMS (cTBS) each had one study. The meta-analysis showed that compared with the control group, LF-rTMS had significant benefits for post-stroke spasticity, and the MAS was significantly reduced (MD: -0.51, 95% CI : −0.78 to −0.24, p < 0.01). However, although other studies had shown certain benefits, they did not reach statistical differ-ences (Figure 2 (B)).

The different stimulation sites of rTMS were divided into different subgroups. Eight of the studies included the unaffected hemispheres of patients with post-stroke spasticity, and the other three studies included the affected hemispheres of patients. The me-ta-analysis showed that compared with the control group, rTMS applied to stimulate the unaffected hemispheres of patients with post-stroke spasticity had significant benefits, and the MAS was significantly reduced (MD: -0.58, 95% CI : −0.80 to −0.36, p < 0.01). However, stimulation of the affected hemispheres also had certain benefits but did not reach statistical differences (Figure 2 (C)).”

“A total of seven RCTs on the effects of tDCS on post-stroke spasticity were included in the study and the measurement outcome for all studies was the MAS. The me-ta-analysis showed that compared with the control group, tDCS had significant benefits for patients with post-stroke spasticity, and the MAS was significantly reduced (MD: -0.65, 95% CI: −1.07 to −0.22, p < 0.01). This meta-analysis showed that there was hetero-geneity between different studies (P < 0.01, I2 = 78 %) (Figure 3 (A)).

The stimulation types of tDCS were divided into different subgroups. Four studies used anodal stimulation and three studies used cathodal stimulation. The meta-analysis showed that compared with the control group, anodal stimulation had significant bene-fits for patients with post-stroke spasticity (MD: -0.74, 95% CI : −1.35 to −0.13, p < 0.05); however, although cathode stimulation also had certain benefits, it did not reach a sta-tistical difference (MD : -0.51, 95% CI : −1.31 to 0.29, p = 0.22) (Figure 3 (B)).

The stimulation intensities of tDCS were divided into different subgroups. There were five studies with stimulation intensity of 2.0 mA, and the other two studies with stimulation intensity of 0.7 mA and 1.2 mA respectively. The meta-analysis showed that compared with the control group, the stimulation intensity of tDCS of 0.7 mA (MD : -1.20, 95% CI : −1.40 to −1.00, p < 0.01) and 1.2 mA (MD : -1.00, 95% CI : −1.26 to −0.74, p < 0.01) had significant effect on patients with post-stroke spasticity. However, the measurement results of other studies had changed but did not reach statistical differences (Figure 3 (C)).”

Changes in the manuscript: Page 9 Line 209-230, Page 10-11 Line 237-256.

  1. In section 3.3, references should be added when mentioning the Cochrane Collaboration recommendations. More justifications are needed to explain how these two authors assessed the risk of bias assessment of 18 included studies.

Response to the comment: Thank you very much for the positive comment. We have modified it in the results section. It reads,

“Risk of bias was assessed using the Cochrane Collaboration recommendations, and the sensitivity results indicated that the results of our meta-analysis appeared to be stable [20].”

Changes in the manuscript: Page 11 Line 267–269.

  1. The information in Fig. 8 and Fig. 9 could be easily combined side by side for a better presentation, since the current design is redundant.

Response to the comment: Thank you very much for suggestions. We have modified it in the results section. We merged figures 8 and 9 in the original manuscript into figure 4 in the revised manuscript. Please see the figure 4.

Changes in the manuscript: Page 12 figure 4.

  1. In the discussion section, more professional perspectives from the authors should be added instead of mentioning only existing literatures studies. In addition, there is no need to separate the study limitations as a subsection 4.1.

Response to the comment: Thank you very much for suggestions. We have modified it in the discussion section. It reads,

“rTMS uses magnetic signals of different frequencies to stimulate the central nervous system in the corresponding parts and relieve limb spasticity in patients after stroke, and to induce brain plasticity and brain network reorganization, promote the rehabilitation of primary and secondary motor cortex [36].”

“Studies have shown that anodal stimulation on the affected side can reduce limb spasticity symptoms in stroke survivors more than cathodal stimulation on the unaffected side [39]. The results of this meta-analysis are consistent with previous studies, which also showed a better effect of anodal tDCS on post-stroke spasticity. However, the mechanism of action of tDCS on post-stroke rehabilitation remains to be further investigated.”

Changes in the manuscript: Page 13 Line 285-289, Line 303-308.

We have removed subsection 4.1 and merged it into the Discussion section. Please see the discussion. It reads,

“Based on this meta-analysis, we also focused on and discussed the results of non-randomized controlled trials of NIBS for post-stroke spasticity. At this stage, no other NIBS have been found in RCTs of patients with post-stroke spasticity, and new techniques still need to be explored in future studies.”

Changes in the manuscript: Page 13 Line 339-342.

  1. Given this paper is a review type, it is not required to add the study limitations in the discussion section. Discussing the pros and cons of reference literature studies can be presented throughout the manuscript.

Response to the comment: Thank you very much for suggestions. We have modified it in the discussion section. We have removed subsection 4.1 and merged it into the Discussion section. Please see the discussion. It reads,

“Based on this meta-analysis, we also focused on and discussed the results of non-randomized controlled trials of NIBS for post-stroke spasticity. At this stage, no other NIBS have been found in RCTs of patients with post-stroke spasticity, and new techniques still need to be explored in future studies.”

Changes in the manuscript: Page 13 Line 339-342.

  1. The first sentence of the conclusion needs to be rewritten because the current review paper did not provide any new results as a research study. In addition, adding the summary of each section in this review would help readers to get a better understanding of the importance of the review.

Response to the comment: Thank you very much for suggestions. We have modified it in the conclusion section. It reads,

“The results of current meta-analysis are encouraging as they suggest that NIBS can promote rehabilitation in patients with post-stroke spasticity.”

Changes in the manuscript: Page 14 Line 344-345.

In addition, we thank the reviewer’s suggestion for summary of each section in the MS. And we modified the wordings at the end of abstract, introduction, as well as the beginning of discussion for this purpose.

Abstract: “This meta-analysis revealed moderate evidence that NIBS reduces spasticity after stroke and may promote recovery in stroke survivors. Future studies of investigating the mechanisms of NIBS on addressing spasticity are warrant to further support the clinical application of NIBS in post-stroke spasticity.”

Introduction: “Currently, the effects of NIBS on post-stroke spasticity are contradictory. Although some studies have reported a beneficial effect of NIBS in the treatment of post-stroke spasticity [11-13], some studies have shown no significant benefit of NIBS in reducing muscle spasticity. A meta-analysis published in 2020 showed no significant effect of rTMS in spasticity management, but it included only five RCTs [14]. Two published meta-analyses on the results of tDCS on post-stroke spasticity also showed some variability without uniform criteria [15, 16]. Therefore, the aim of this study is to conduct a systematic review and meta-analysis of the effectiveness of NIBS in the management of spasticity in patients after stroke.”

Discussion: “In this current study, a meta-analysis of the effect of NIBS on spasticity for post-stroke populations was performed and it included 18 RCTs, the most relevant RCTs to date based on stringent inclusion and exclusion criteria. The results of the meta-analysis proved that NIBS has a positive effect on post-stroke spasticity. In addition, the sub-group of NIBS (i.e. tDCS and TMS) on post-stroke spasticity was also conducted.”

“Based on this meta-analysis, we also focused on and discussed the results of non-randomized controlled trials of NIBS for post-stroke spasticity. At this stage, no other NIBS have been found in RCTs of patients with post-stroke spasticity, and new techniques still need to be explored in future studies.”

  1. Various information is missing or unnecessary information in the reference, for example, the number in [1][4], '-' at the end of [2], the missing page number in [40], and so on. The authors need to check each entry carefully to ensure all information is correct.

Response to the comment: Thank you for the comment. We have modified it in the references section accordingly and also checked other references.

Changes in the manuscript: Page 13-15 Reference.

  1. The journal or conference names in the references need to be consistent throughout the manuscript, either all with full names or abbreviated names.

Response to the comment: Thank you for the comment. We have checked all references for accuracy accordingly and also checked other references.

Changes in the manuscript: Page 13-15 Reference.